# In Vitro Evaluation of Phytobiotic Mixture Antibacterial Potential against *Enterococcus* spp. Strains Isolated from Broiler Chicken

**DOI:** 10.3390/ijms25094797

**Published:** 2024-04-27

**Authors:** Karolina Wódz, Karolina A. Chodkowska, Hubert Iwiński, Henryk Różański, Jakub Wojciechowski

**Affiliations:** 1Laboratory of Molecular Biology, Vet-Lab Brudzew, Turkowska 58c, 62-720 Brudzew, Poland; jakub.wojciechowski@labbrudzew.pl; 2Ferma Podolany Spółka z o.o., ul. Zakładowa 7, 26-670 Pionki, Poland; 3AdiFeed Sp. z o.o., Chrzanowska 15, 05-825 Grodzisk Mazowiecki, Poland; hubert.iwinski@adifeed.pl (H.I.); rozanski@rozanski.ch (H.R.); 4Laboratory of Industrial and Experimental Biology, Institute for Health and Economics, Carpathian State College in Krosno, Rynek 1, 38-400 Krosno, Poland

**Keywords:** *Enterococcus* spp., broiler, osteomyelitis, resistance genes, *E. cecorum*, phytobiotics

## Abstract

*Enterococcus* spp. are normal intestinal tract microflorae found in poultry. However, the last decades have shown that several species, e.g., *Enterococcus cecorum*, have become emerging pathogens in broilers and may cause numerous losses in flocks. In this study, two combinations (H1 and H2) of menthol, 1,8-cineol, linalool, methyl salicylate, γ-terpinene, p-cymene, *trans*-anethole, terpinen-4-ol and thymol were used in an in vitro model, analyzing its effectiveness against the strains *E. cecorum*, *E. faecalis*, *E. faecium*, *E. hirae* and *E. gallinarum* isolated from broiler chickens from industrial farms. To identify the isolated strains classical microbiological methods and VITEK 2 GP cards were used. Moreover for *E. cecorum* a PCR test was used.. Antibiotic sensitivity (MIC) tests were performed for all the strains. For the composition H1, the effective dilution for *E. cecorum* and *E. hirae* strains was 1:512, and for *E. faecalis*, *E. faecium* and *E. gallinarum*, 1:1024. The second mixture (H2) showed very similar results with an effectiveness at 1:512 for *E. cecorum* and *E. hirae* and 1:1024 for *E. faecalis*, *E. faecium* and *E. gallinarum*. The presented results suggest that the proposed composition is effective against selected strains of *Enterococcus* in an in vitro model, and its effect is comparable to classical antibiotics used to treat this pathogen in poultry. This may suggest that this product may also be effective in vivo and provide effective support in the management of enterococcosis in broiler chickens.

## 1. Introduction

*Enterococci* are Gram-positive facultative anaerobic bacteria that are part of a natural intestinal microbiota in humans and animals [1,2]. There are more than 40 recognized species classified in the *Enterococcus* genus. In humans, *Enterococcus faecium* and *Enterococcus faecalis* are recognized as the third- and fourth-most prevalent human pathogens in the world [3] and are responsible for 11–13% of all bacteremia cases in Europe and North America. In addition, 12% of all nosocomial infections are associated with this group of bacteria [4]. What seems interesting is that a genetic link has been found between the *E. faecalis* in the human urinary tract that is causing its infection and those isolated from poultry. This, in turn, indicates that some of the infections in humans may be classic zoonosis, and poultry itself is a significant source of this dangerous germ [5,6,7].

In poultry, this group of bacteria is linked to several health problems, among which endocarditis, septicemia and enterococcal spondylitis (ES) seem to be the most important, from a veterinary and also economical point of view. It should be emphasized that different strains of *Enterococci* may cause the different symptoms described above. The most common strains in poultry are *E. feacalis*, *E. faecium*, *E. gallinarum*, *E. hirae* and *E. cecorum*. *E*. *faecalis*, most common in poultry, is identified as among the most relevant antimicrobial-resistant (AMR) bacterium in the EU for poultry [8], being responsible for endocarditis [9,10,11], hepatic granulomas in turkeys [12] and arthritis and amyloidosis both in layers [13] and broiler breeders [14,15]. Moreover, it causes septicemia, salpingitis and peritonitis [13,16], growth depression and poor flock uniformity [17], pulmonary hypertension syndrome [18] and amyloid arthropathy [14,19]. *E. faecium* is related to the acute septicemia of white Pekin ducklings [20], but in broilers, it has been widely used as a probiotic [21] as a part of gut microbiota [22]. However, in broilers, it is also associated with endocarditis, septicemia, amyloid arthropathy and spondylitis and musculoskeletal diseases [23]. It is also one of the most multidrug resistant bacteria in the *Enterococcus* group [24]. *Enterococcus gallinarum* and *Enterococcus hirae* were isolated from flocks where massive lameness was observed [25]. Moreover, the second strain was previously reported in young chickens causing focal necrosis of the brain, osteomyelitis and endocarditis [26,27]. For many years, *E. cecorum* in poultry was recognized as a classic intestinal flora, a commensal organism producing lactic acid. However, since 2002, there have been local epidemics of osteomyelitis due to the pathogenic strains of *E. cecorum* occurring in broiler and broiler breeder flocks; scientist began to look closely at not only the disease itself but, above all, *E. cecorum* and the entire group of *Enterococcus* found in poultry. It is not entirely clear (due to the lack of complete data) if there is a risk to humans from poultry products from flocks where the presence of bacteria belonging to the *Enterococcus* spp. group has been confirmed. Nevertheless, there are other, equally important aspects, such as the transmission of drug-resistant enterococci with poultry products. The increasing drug resistance of these bacteria is related to the transfer of plasmids and transposons and chromosomal exchange or mutation, which makes it difficult to effectively manage infected herds and treat diseases caused by *Enterococcus* spp.

Due to growing antibiotic resistance but also the general trend of reducing antibiotics in both human and animal medicine, more and more attention is being paid to the possibility of using effective alternatives to classical antibiotic therapy with a high margin of safety. The mentioned safety issue concerns not only the lack of adverse effects and the lack of toxicity for the target species in which the product will be used but, in the case of animals whose meat and eggs end up as human food, the lack of risk of residues of dangerous substances. For this reason, preparations with pro- and prebiotics, and their combinations, but also phytobiotics are becoming more and more popular. A number of previous studies, also in poultry, clearly indicate that preparations from this group can not only effectively support the overall health of a flock, improving the balance of intestinal bacterial flora, but also stimulate immunity and help in managing the occurrence of protozoal, viral and bacterial diseases in broiler flocks. Essential oils are characterized by very good antimicrobial properties, including *Enterococci* strains. This is due to the presence of many bacteriostatic and antibacterial substances in their composition, e.g., lactones, esters, terpenes, terpenoids, biopolymers, ethers, etc. The variability in essential oils and their constituents is tremendous. It also depends on many factors, both abiotic (soil, sunlight exposure, humidity) and biotic (drying, storage, agrotechnical methods). The composition also depends on the species. These factors have a significant effect on an essential oil’s composition and properties [28]. It has been shown in several scientific papers that a mixture of some essential oils or their constituents may exhibit either a synergistic or antagonistic effect. Synergism has been observed in the different combinations of the essential oils with, among others, essential oils or their components [29,30,31], antibiotics, metal ions or organic acids [32].

Earlier scientific works, both on in vitro models [33,34], indicate the antibacterial effect of a number of phytobiotic mixtures, while maintaining their safety of use in animals [35,36]. In previous studies, conducted on the same blend of phytobiotics, on an in vitro model with field strains of *Salmonella* spp., *E. coli* (including APEC), the tested mixture demonstrated also strong antibacterial effectiveness.. The presented work complements the analyses with an assessment of the antibacterial activity against the group of *Enterococcus* spp., which in recent years, next to *E. coli* and bacteria from the *Clostridium* spp. group, is one of the greatest challenges for broiler chicken breeding, causing losses at various stages of rearing.

So far, there have been few studies of the use of mixtures of phytobiotics and metal chelates that have analyzed their antibacterial effects. It should be noted that most research has focused on single active substances or their simple mixtures. Moreover, the vast majority of in vitro studies have been conducted on classic reference strains of bacteria, not those isolated from living organisms. Reference strains often show a different drug resistance to field strains. Our analysis using field strains not only provides information on the effects of the tested phytoncide mixture but also shows the current drug resistance of various field strains of *Enterococcus* spp. on broiler chicken farms in Poland. The described observations are part of a project in which phytobiotic mixtures previously tested in vitro will be tested on in vivo models, first on a small scale and then on industrial broiler farms. The doses indicated in the presented study will constitute a starting point for further research on the development of effective solutions to support classical antibiotic therapy and/or its alternatives.

## 2. Results

### 2.1. Post-Mortem Examination

The appearance of initial clinical symptoms after the second week of age was mostly related to problems with movement, asymmetry of the legs and lameness, as well as increasing mortality, and required necropsy examinations of dead and selected birds. The most common autopsy lesions in birds in which *Enterococcus* spp. were confirmed included the following:Arthritis;Femoral head necrosis (one or both sides);Necrosis within the vertebrae;Necrotic changes in the pelvic bones;Septicemia;Endocarditis;Peritonitis;Pericarditis/hydropericardium.

Moreover, in herds where clinical symptoms of *E*. *cecorum* were observed and confirmed by post-mortem and laboratory tests, the following were found:-poor flock uniformity in terms of slaughter weight;-severe changes in the skin of the soles (pododermatitis) and ankle joints (hock burns);-brittle bones in the hip joints;-general decrease in the quality of the raw material.

### 2.2. Biochemical Features 

A total of 694 *Enterococus* spp. strains from day-old chicks and from broilers were isolated. A total of 694 *Enterococcus* isolates were obtained, representing five species: *E. cecorum* (n = 154; prevalence 22.19%), *E. faecalis* (n = 453; prevalence 65.27%), *E. faecium* (n = 1; prevalence 0.14%), *E. gallinarum* (n = 57; prevalence 8.21%) and *E. hirae* (n = 29; prevalence 4.19%).

The accuracy of the identification obtained with the first biochemical test, VITEK 2 COMPACT and GP cards, for some of the isolates of *Enterococcus* colonies on Columbia and CNA agars was insufficient. Biochemical identification was inconclusive, especially for *E. cecorum* and *E. columbae* (Table 1). Finally, *E. cecorum* was identified based on the results obtained using real-time PCR.

### 2.3. Phytoncide Mixture Test

The analysis of the two phytobiotic compositions revealed considerable antibacterial properties. The minimum inhibitory concentration (MIC) and minimum bactericidal concentration (MBC) values fell within the range 1:512 to 1:1024. For the composition H1, the effective dilution for the *E. cecorum* and *E. hirae* strains was 1:512, while for the *E. faecalis*, *E. faecium* and *E. gallinarum* strains, the effective dilution was 1:1024.

The second mixture (H2) showed very similar results with an effectiveness at 1:512 for *E. cecorum* and *E. hirae* and 1:1024 for *E. faecalis*, *E. faecium* and *E. gallinarum*.

### 2.4. Antibiotic Susceptibility Testing

A total of 694 *Enterococus* spp. strains from one-day-old chickens and from broilers were isolated. A total of 154 *E. cecorum*, 453 *E. faecalis*, 1 *E. faecium*, 57 *E. gallinarum* and 29 *E. hirae* strains were tested for antimicrobial resistance against a panel of 25 antimicrobials.

All *Enterococcus* spp. (*E. cecorum*, *E. faecalis*, *E. faecium*, *E. gallinarum* and *E. hirae*) strains were susceptible to amoxicillin and amoxicillin with clavulanic acid. Detailed results are shown in Figure 1. A resistance to vancomycin was not detected in all tested *E. cecorum*, *E. faecalis*, *E. faecium* and *E. hirae* spp. strains. Moreover, all the *Enterococcus* strains were intrinsically resistant to cephalosporins (cephalexin, cephapirin and cefquinome).

The antimicrobial resistance in the isolates of *Enterococcus* spp. is presented in Table 2.

According to TECOFF, *E. faecium* and all *E. faecalis* and *E. hirae* strains showed resistance to gentamicin (MIC > 32), neomycin (MIC > 128) and streptomycin (MIC > 128). Ten *E. faecalis* strains (2.21%) were resistant to florfenicol (MIC > 8). A total of 42.83% of *E. faecalis* strains were resistant to norfloxacin, and 57.17% were intermediate. For erythromycin, 100% of *E. faecalis* strains were resistant, whereas for *E. cecorum*, 96.1%. This indicates the risk of the further development and spread of the phenotypic resistance of the most common enterococci to these antimicrobials.

## 3. Discussion

The presented data indicate that both the mixtures of phytoncides designated H1 and H2 have effective antibacterial activity for the analyzed strains of *Enterococcus* spp. in an in vitro model. While the problem of *Enterococcus* spp. and its classic therapy in poultry has been widely analyzed, the potential use of phytobiotics in the control of infections of this type is a relatively new issue. It should be emphasized that the presented work is another study of this particular mixture confirming its in vitro antibacterial effectiveness in combating bacteria, which currently constitute one of the greatest challenges in the poultry industry. Two previous studies showed that the H1 and H2 mixture (MIC 1:512 and 1:1024) were effective against *E. coli* (including APEC) [33] and against selected strains of *Salmonella spp*. (MIC 1:256) [34]. Here, we observed that the minimum inhibitory concentration (MIC) and minimum bactericidal concentration (MBC) values fell within the range 1:512 to 1:1024. For the composition H1, the effective dilution for the *E. cecorum* and *E. hirae* strains was 1:512, while for the *E. faecalis*, *E. faecium* and *E. gallinarum* strains, the effective dilution was 1:1024. The second mixture (H2) showed very similar results with an effectiveness at 1:512 for *E. cecorum* and *E. hirae* and 1:1024 for *E. faecalis*, *E. faecium* and *E. gallinarum*. In this study, mixtures of essential oils, organic acids and metal ions were used. The combination effect on their antibacterial properties and MOA can be different than pure essential oils. We assumed the expected MOA of the used mixtures based on their constituents’ antibacterial properties. The antibacterial properties mode of action of used essential oils mostly based on the disturbance of cell membrane permeability. The consequences of that are leakage of cell content, pore formation or vesiculation in the cell membrane. Moreovermetal ions are transported to the cell by different mechanisms than essential oils. They are transferred passively by ion channels. Due to the two different MOAs, we assume that they have a synergistic effect, thus allowing them to facilitate each other’s penetration of the cell membrane and effectiveness. This indicates that both mixtures may be potentially effective against infections that come from hatchery or parent flocks, most often observed in one-day-old chicks (single or combined; *E. coli* and *Enterococcus* spp.) as well as in further stages of rearing, where both groups of bacteria cause significant losses. 

The occurrence of *Enterococcus* spp. itself is consistent with what was observed a few years ago in Poland [37,38]. It should be emphasized that in previous studies, extremely sensitive PCR tests were used for the detection of *E. cecorum* in day-old chicks. A total of 694 *Enterococcus* belonging to five species were detected in our samples: *E. cecorum* (n = 154; prevalence 22.19%), *E. faecalis* (n = 453; prevalence 65.27%), *E. faecium* (n = 1; prevalence 0.14%), *E. gallinarum* (n = 57; prevalence 8.21%) and *E. hirae* (n = 29; prevalence 4.19%). It should be emphasized that no strains of *E. cecorum* were found in 1-day-old chicks. Bzdil et al. [39] also detected in 1-day-old chicks four species of enterococci, *E. faecalis*, *E. faecium*, *E. gallinarum* and *E. hirae*, but not *E. cecorum*. The results regarding the drug resistance of the analyzed strains of *Enterococcus* spp. are mostly consistent with what the abovementioned authors observed a few years ago. The differences are probably due to the fact that in previous studies the sampling period was longer, and the number of analyzed samples was larger and covered a larger geographical range. In addition, the authors also analyzed samples from laying flocks, parent flocks and turkeys. However, both previous research and ours clearly indicate that the problem related to the presence of various strains of *Enterococcus* spp. and their drug resistance in poultry production is increasing. 

In our study, 42.83% of *E. faecalis* strains were resistant to norfloxacin, and 57.17% were intermediate. For erythromycin, 100% of *E. faecalis* were resistant, whereas for *E. cecorum*, 96.1%. This indicates the risk of the further development and spread of the phenotypic resistance of the most common enterococci to these antimicrobials. In other studies, resistance to erythromycin in chickens varied between 31.1% and 61.0% [40] or 40% and 73.1% [39]. It is noteworthy that our findings regarding the *E. faecium* strain revealed 100% susceptibility to the tested antimicrobials. However, it is important to acknowledge the limited sample size of only one tested strain. Analyzing the obtained results in the category of food safety, their compliance with the results related to the occurrence of various strains of *Enterococcus* spp., and their drug resistance in poultry production around the world is visible [41,42,43,44]. Previous publications clearly indicate that the problem of the presence of this type of bacteria in food of animal origin is increasing and poses a high risk to consumers [43]. Hence, the possibility of using effective measures to eliminate this type of bacteria at the poultry farm level seems to be a key element of prevention and food safety.

As already emphasized, there are few studies analyzing the antibacterial effect of phytobiotic products on the most common *Enterococcus* strains in broilers (reference strains and strains occurring in industrial breeding) [45,46]. The problem of comparing results probably stems from the fact that the mixture of phytobiotics used in the experiment has a unique composition, not found in any other preparation. Available works most often analyze the impact of one substance or products with a much simpler composition than the one analyzed in our study. What additionally distinguishes the presented study is the fact that the mixture of phytobiotics used included chelates of metals and organic acids. These ingredients have a chance to effectively enhance the bactericidal effect of the preparation compared to the phytobiotic mixtures described so far.

Our results are consistent with previous observations and trends in the occurrence of various strains of *Enterococcus* spp. in commercial broiler flocks. The clinical and autopsy picture (Figure 2) confirmed by microbiological tests indicates that both strains isolated at the stage of day-old chicks and those from older birds may cause a negative impact on the entire broiler flock rearing and, ultimately, on the quality of the final product, i.e., the carcass. The described anatomical changes, observed many times during rearing, despite the introduction of effective treatment to reduce mortality and selection, were already observed at the stage of post-mortem examination during thinning and final depopulation (Figure 3 and Figure 4).

In *E. faecalis*, susceptibility to amoxicillin with and without a beta-lactamase inhibitor is the expected phenotype, while in *E. faecium*, resistance is common. 

However, the use of single active substances entails the risk of pathogen resistance developing more rapidly, as it has occurred with some antibiotics. Therefore, it was decided to use a mixture of phytoncides with the highest antibacterial potential, especially against *Enterococcus*. The use of a mixture of active substances had been already shown in previous results obtained against *Salmonella* spp. and *E. coli* strains [33,34]. The synergistic and antagonistic effect of combinations of different terpenes against bacteria was presented by Gallucci et al. [29]. They showed the stronger effect of a mixture of menthol and thymol than the individual components. Similar reports can also be found on the combinations of other phytoncides investigated in this study, e.g., thymol and p-cymene, linalool, p-cymene and γ-terpinene, thymol, menthol, eucalyptol and methyl salicylate [47]. Moreover, Pecarski et al. [48] evaluated the chemical composition and antimicrobial activity of essential oils of thyme and oregano where one of the tested bacteria was *Enterococcus faecalis* ATCC 25929. The efficacy of oregano and thyme essential oils in the inhibition of *Enterococcus faecalis* was evaluated at varying concentrations, namely, 1, 3, 5 and 10 µL/mL, which proved to be sufficient to achieve a significant reduction in microbial growth. The minimum inhibitory concentration (MIC) value for thyme was 1 µL/mL. In contrast, oregano demonstrated bacteriostatic activity with a MIC value of 3 µL/mL. The same authors evaluated the antimicrobial activity of the essential oil of cumin (limonene, p-cymene, α-pinene and eucalyptol) and fennel (*trans*-anethole, l-fenchone, estragole, limonene and α-pinene) [48]. They noticed that *E. faecalis* was resistant to both essential oils, as well as Gram-negative bacteria *E. coli* that was resistant to the fennel essential oil activity. These observations are completely different from those obtained by us, which may indicate that the unique mixture of phytobiotics and its effectiveness is based on the phenomenon of synergy. Additionally, the differences may result from the fact that in our study we observed a composition of phytobiotics and not individual herbs. Regarding *E. cecorum*, the available literature is still limited. However, in vitro studies have demonstrated antimicrobial properties for carvacrol and mixtures of medium-chain fatty acids, including hexanoic, octanoic, decanoic and dodecanoic acid [49].

Among the *Enterococcus*, strains identified in this study, the *E. cecorum* seems to be the most interesting due to the fact that previous works have indicated not only its increasingly frequent appearance in broiler flocks but also increasing drug resistance. As has already been emphasized, this strain not only poses a significant threat to the birds themselves and reduces the quality of the raw material (lack of flock uniformity, changes in the carcass and organs) but may also pose a threat to consumers. The clinical symptoms of *E. cecorum* in the analyzed flocks were most often observed in the 2–6th week of life, with isolated cases of single birds with symptoms of lameness and necrosis of the femoral head. This is consistent with previous studies [50,51,52]. Many authors indicate that *E. cecorum* is highly sensitive to amoxicillin, and this antibiotic is most often used in clinical practice on industrial poultry farms. It should be emphasized that in the case of the treatment of older herds, in addition to the criterion of effectiveness, another important criterion for choosing therapy is its cost and the antibiotic withdrawal period. In Polish conditions, amoxicillin meets all of the above. It should be noticed that a large percentage of samples from one-day-old chicks showed a positive result in PCR analysis for the presence of *E. cecorum*. This result should be combined with the fact that there is a very high probability that this bacterium is present in the parent flocks from which the chicks came or in the hatchery environment. The problem of *E. cecorum* in broiler parent flocks has been widely analyzed in recent years, especially in terms of embryo death and transmission of the disease to offspring [53,54,55]. Both the treatment of parent flocks and broiler flocks at a later stage (e.g., near depopulation) poses many problems (economics, withdrawal period, risk of antibiotic residues in meat). Similar problems related to the occurrence of *Enterococcus* spp. infections and its treatment apply to laying hens [38,56]. In the case of this type of poultry production, the issue of withdrawal period, the danger of drug residues in eggs intended for human consumption but also the increasing drug resistance of the mentioned strains deserve special attention [39,56,57]. It seems that effective phytobiotic products that do not have a withdrawal period have a chance to become permanent in the prevention and control of infections with various strains of *Enterococcus* spp. However, the presented study concerns an in vitro model and is a pilot study. For this reason, further investigations should be conducted to test the two analyzed phytobiotic mixtures on a larger scale and develop the most effective drug administration regimens. The schemes should take into account not only differences in production groups (laying hen, parent flock, broiler) but also effective doses, time and the moment of the administration of the products. 

## 4. Materials and Methods

### 4.1. Broiler Sampling

Samples were routinely collected from industrial broiler farms in Poland over 24 months (April 2021–April 2023). Samples were taken from both one-day-old chicks (from the hatchery car cages during the delivery to the farms, to be sure that there was no possibility for sampling contamination) and older birds from flocks which presented the clinical symptoms related to lameness, moving problems, higher mortality/selection, lower uniformity, lower weight and decreasing feed and water intake. The second group consisted of both birds that died recently and live birds that already showed clinical signs (lameness, mobility problems, unnatural leg position) or came from a flock suspected of being at risk of the disease. The transport and autopsy procedures were performed as previously described. Samples from organs, single tissues, were collected and prepared for further analysis based on an internal laboratory protocol.

### 4.2. Enterococcus spp. Isolation and Identification

*Enterococcus* spp. (*E. cecorum*, *E. faecalis*, *E. faecium*, *E. gallinarum* and *E. hirae*) strains were isolated from broiler liver and spleen, femur head, joints and tendon sheath and vertebral column. Organ sampling was performed during necropsy according to an internal laboratory protocol. All samples were cultured onto Columbia Agar with 5% Sheep Blood CNA Agar (both from Graso, Starogard Gdański, Poland). Plates were then incubated at 37 °C for 24 h under aerobic conditions. *Enterococcus* isolates were initially identified based on colony morphology, type of hemolysis and catalase reaction. *Enterococcus* Selective Agar BAA (Bile Aesculin Azide Agar, ThermoScientific) was used for the presumptive identification of *Enterococcus* spp. and differentiation from *Streptococcus* spp. based on esculin hydrolysis. Isolated strains were then identified with VITEK 2 Compact and GP cards (Biomerieux, Craponne, France). A real-time PCR method based on the detection of genes specific for *E. cecorum* (Kylt, Emstek, Germany) was used to differentiate *E. cecorum* from *E. columbae*. DNA was extracted from bacterial cells using an automated magnetic isolation method (MagnifiQ™ Pathogen kit, A&A Biotechnology, Gdansk, Poland) and Nucleic Acid Purification System—Auto-Pure 96 (Hangzhou Allsheng Instruments, Wuxi, China) and real-time PCR using Applied 7500FAST (ThermoFisher, Waltham, MA, USA). PCR conditions: 95 °C for 10 min., 42 cycles: 95 °C for 15 s and 60 °C for 1 min. Fluorescence detection in channels FAM (target) and HEX (internal control). A schematic illustration of the steps carried out in the study is presented in Figure 5.

### 4.3. Phytoncide Combination

The tested combination consisted of equivalent amounts of common phytoncides menthol, 1,8-cineol, linalool, methyl salicylate, γ-terpinene, p-cymene, *trans*-anethole, terpinen-4-ol and thymol. Compound purity was at minimum ≥ 95%, and all were purchased from Sigma-Aldrich (St. Louis, MO, USA). The phytoncide combination was mixed, heated and left overnight. After that, Polysorbate 80 was added as an emulsifying agent, to allow easier dissolution in culture media. Stock solutions of the bioactive compounds were diluted in fresh sterile Mueller Hinton broth cation adjusted (Graso, Gdansk, Poland) to reach the final concentrations tested.

### 4.4. Phytoncide Mixture Test by Broth Microdilution Method

The assessment of the impact of the phytoncide mixtures on *Enterococcus* spp. (*E. cecorum*, *E. faecalis*, *E. faecium*, *E. gallinarum*, *E. hirae*) was performed using the broth microdilution method according to the ISO 20776-1:2006 [58]. Two-fold serial dilutions of the phytoncide mixture were prepared in sterile Mueller Hinton (M-H) broth with cation to achieve a final volume of 2 mL per vial. Inoculum was prepared in a sterile 0.9% saline solution, derived from the overnight culture isolated on sheep blood agar, with turbidity adjusted to 0.5 McFarland standard. The bacterial suspensions were then diluted one hundred times in M-H broth to obtain an inoculum of 10^6^ CFU/mL. Next, 1 mL of bacterial inoculum was transferred to vials containing 1 mL of the diluted mixture. The following test dilutions were prepared per row: 4, 8, 16, 32, 64, 128, 256, 512, 1024, 2048, 4096 and 8192. Vials containing 1mL M-H broth only, without product, and 1ml inoculum were used as positive controls. Vials containing only 1ml dilutions and 1ml M-H broth were used as negative controls. Vials were then incubated at 35 ± 1 °C for 21 ± 3 h. After incubation, the lowest concentration of the product which completely inhibited visible growth was assessed. After inoculation of the vials, bacterial suspensions in saline were streaked on Columbia Agar with 5% sheep blood agar for purity testing. After an overnight incubation at 37 °C, the cultures were examined for the presence of morphologically characteristic colonies.

### 4.5. Antibiotic Susceptibility Testing

Susceptibility to antibiotic substances was performed using the minimal inhibitory concentrations using a broth microdilution method in 96-well MICRONAUT Special Plates with antimicrobials: β-lactams/aminopenicillin (amoxicillin—AMX, amoxicillin and clavulanic acid—AMX/CL), β-lactams/I generation cephalosporins (cephalexin—CFX, cephapirin—CPH), β-lactams/III generation cephalosporins (ceftiofur—CFTI), β-lactams/IV generation cephalosporins (cefquinome—CFQ), β-lactams/penicillin (cloxacillin—CLO, penicillin G—PG, nafcillin—NAF), aminoglycoside (gentamicin—GEN, neomycin—NEO, streptomycin—STR), polymyxins (colistin—COL), fluoroquinolones (enrofloxacin—ENR, norfloxacin—NOR), tetracyclines (doxycycline—DOX, oxytetracycline—OXY), macrolides (erythromycin—ERY, tylosin—TYL), florfenicol—FLR, lincosamides (lincomycin—LIN, lincomycin/spectinomycin—LIN/SP), trimethoprim-sulfamethoxazole—TR/SMX, tiamulin—TIA, tylvalosin—TYLV (MERLIN Diagnostika GmbH, Bremen, Germany). The MICs were interpreted according to the Clinical and Laboratory Standards Institute (CLSI M 100-ED 30) and EUCAST 2024 breakpoints (breakpoint Table 14.0). The set of antimicrobials (Table 3 and Table 4) was selected in order to reflect their importance both for human and veterinary medicine (EUCAST 14.0, CLSI_VET, 2020; CLSIM, 2020). In addition, tentative epidemiological cut-off values (TECOFFs) were used for antibiotics without breakpoints for *Enterococcus* spp. (EUCAST 2024) [49], Table 5.

Vancomycin resistance was assessed by Kirby–Bauer disk diffusion with a 5 µg/mL disk (Graso) according to the CLSI and EUCAST guidelines. Vancomycin resistance was defined as ≤12 mm. CHROMagar VRE (Graso, Gdansk, Poland) was used for the detection of Van A/Van B *Enterococcus*. Vancomycin resistance is the expected phenotype for *E. gallinarum*, and therefore susceptibility testing was not performed.

## 5. Conclusions

To our knowledge, this is the first study which demonstrates the in vitro effectiveness of a mixture of phytobiotics containing menthol, 1,8-cineol, linalool, methyl salicylate, γ-terpinene, p-cymene, *trans*-anethole, terpinen-4-ol and thymol on field strains of *Enterococcus* isolated from the tissues of broiler chickens kept on industrial farms. This study is part of a scientific project on alternatives to classic antibiotics used to treat the most common bacterial infections in broiler chickens. In previous studies on the same and very similar mixtures of phytobiotics, we demonstrated their effectiveness in vitro against selected field strains of *Salmonella* spp., *E.coli* (including APEC) [33,34], as well as in a in vivo model, that these products, even in high doses, are safe for broiler chickens. Additionally, unlike antibiotics, there is no risk of residues in tissues and no withdrawal period. The presented study is an introduction to currently conducted research on in vivo models (within the so-called university and field trial), where it will be possible to confirm the effectiveness of the above composition and indicate therapeutically effective doses. The above study clearly indicates that the problem of *E. cecorum* and its drug resistance is increasingly present in broiler flocks and may be primarily related to vertical infection from parent flocks (numerous positive results in tests from one-day-old chicks). This, in turn, indicates the need for the regular monitoring of both parental flocks and broilers during the rearing stage, as well as the development of effective methods for the prevention and treatment of this type of infection. It seems that the proposed composition of phytobiotics may be an effective support for classical therapy or a preventive solution in the control of *E. cecorum* infections.

## Figures and Tables

**Figure 1 ijms-25-04797-f001:**
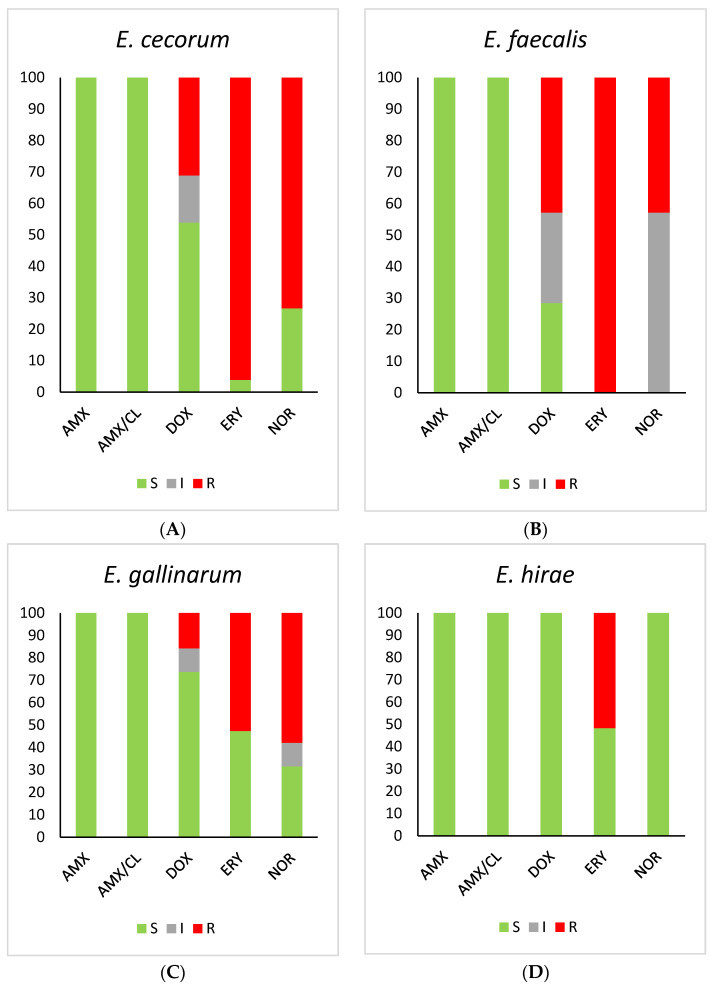
Percentage of susceptible, intermediate and resistant isolates of (**A**) *E. cecorum* (n = 154), (**B**) *E. faecalis* (n = 453), (**C**) *E. gallianrum* (n = 57) and (**D**) *E. hirae* (n = 29). AMX—Amoxicillin, AMX/CL—Amoxicillin–clavulanic acid, DOX—Doxycycline, ERY—Erythromycin, NOR—Norfloxacin; R—resistant, I—intermediate, S—susceptible.

**Figure 2 ijms-25-04797-f002:**
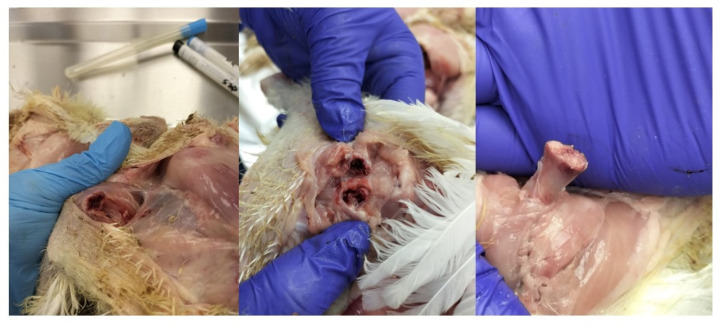
The most frequently observed post-mortem lesions associated with *E. cecorum* infection in broiler chickens are related to necrotic changes in the femoral head. Clinically, it manifests itself as unilateral and/or bilateral lameness. The lack of movement and, consequently, the lack of water and feed intake results in lower weight gain, worse overall health status and decreased uniformity of the flock.

**Figure 3 ijms-25-04797-f003:**
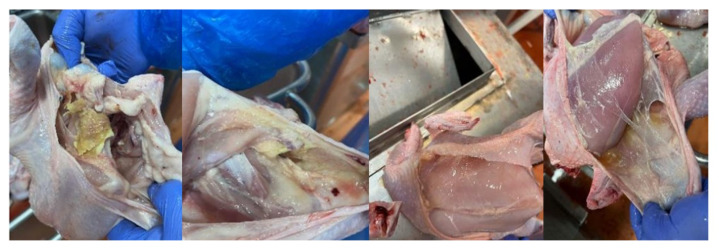
During post-mortem evaluation at the slaughterhouse, a large percentage of flocks with clinical symptoms of *E. cecorum* infection, despite treatment, showed a number of changes in the carcass that reduced its quality. A large group of changes was related to the inability to move properly (lameness), being trapped and trampled by other birds (bedsores, hematomas, abscesses, skin scratches, etc.).

**Figure 4 ijms-25-04797-f004:**
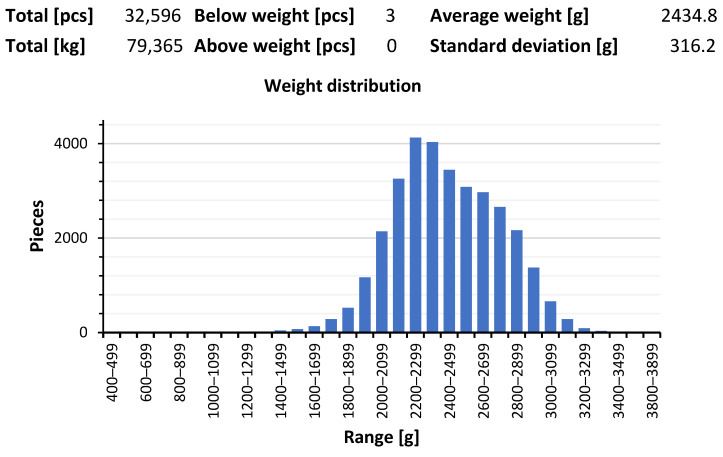
A common post-mortem observation was the large variation in flock weights (low uniformity) that accompanied birds after *E. cecorum* infection after the 20th day of life.

**Figure 5 ijms-25-04797-f005:**
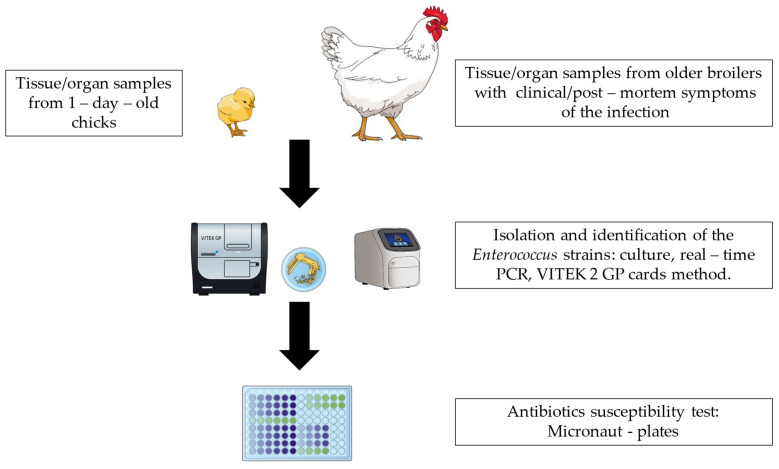
Schematic illustration of the steps carried out in this study.

**Table 1 ijms-25-04797-t001:** Summary comparison of selected biochemical features of *E. cecorum and E. columbae*.

Test	*E. cecorum*	*E. columbae*	*E. cecorum*/*E. columbae*
D-xylose	−	+	+
beta-galactosidase	−/+	+	+
cyclodextrin	+	−/+	+
beta-glucuronidase	−/+	−	−
alanine arylamidase	−/+	−	−
D-ribose	−/+	+	+
lactose	−/+	+	+
bacitracin resistance	−/+	+	+
growth in 6.5% NaCl	−/+	−	+
mannitol	−/+	+	−

“+”— positive results, “–“ — negative results, ”−/+” — variable results.

**Table 2 ijms-25-04797-t002:** Summary comparison of the phenotypic AMR results from 694 *Enterococcus* isolates.

Antibiotic	MIC Breakpoints µg/mL (CLSI, EUCAST)				
	*E. cecorum*	*E. faecalis*	*E. faecium*	*E. gallinarum*	*E. hirae*
Amoxicillin	S (154/154)	S (453/453)	S (1/1)	S (57/57)	S (29/29)
Amoxicillin–clavulanic acid	S (154/154)	S (453/453)	S (1/1)	S (57/57)	S (29/29)
Doxycycline	S (83/154) I (23/154) R (48/154)	S (129/453) I (130/453) R (194/154)	S (1/1) I (0/1) R (0/1)	S (42/57) I (6/57) R (9/57)	S (29/29) I (0/29) R (0/29)
Erythromycin	S (6/154) I (0/154) R (148/154)	S (0/453) I (0/453) R (453/453)	S (1/1) I (0/1) R (0/1)	S (27/57) I (0/57) R (30/57)	S (14/29) I (0/29) R (15/29)
Norfloxacin	S (41/154) I (0/154) R (113/154)	S (0/453) I (259/453) R (194/453)	S (1/1) I (0/1) R (0/1)	S (18/57) I (6/57) R (33/57)	S (29/29) I (0/29) R (0/29

S—Susceptible, I—Intermediate, R—Resistant.

**Table 3 ijms-25-04797-t003:** Breakpoints CLSI M100 for *Enterococcus* spp.

Antibiotics	Interpretative Category and MIC Breakpoints µg/mL		
	S	I	R
Erythromycin	≤0.5	1–4	≥1
Doxycycline	≤4	8	≥16
Norfloxacin	≤4	8	≥16

**Table 4 ijms-25-04797-t004:** Breakpoints EUCAST for *Enterococcus* spp.

Antibiotics	Interpretative Category and MIC Breakpoints mg/L	
	S	R
Amoxicillin	≤4	>8
Amoxicillin–clavulanic acid	≤4	>8

**Table 5 ijms-25-04797-t005:** TECOFFs (tentative epidemiological cut-off values) according to EUCAST.

Antibiotic	Tentative MIC Breakpoints		
	*E. faecalis*	*E. faecium*	*E. hirae*
Doxycycline	1	0.5	-
Erythromycin	4	4	1
Florfenicol	8	8	-
Gentamicin	64	32	32
Neomycin	256	64	128
Streptomycin	512	128	128

## Data Availability

Dataset available on request from the authors.

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
