# Peer review of "In Vitro Evaluation of Phytobiotic Mixture Antibacterial Potential against Enterococcus spp. Strains Isolated from Broiler Chicken"

_ijms, 2024, doi:10.3390/ijms25094797_

Round 1
Reviewer 1 Report
Comments and Suggestions for Authors
This study is interesting and show important findings within the antibacterial strategy field; however, some concerns should be considered as follows:
1. Title: should be antibacterial not antimicrobial since the investigations were conducted on bacterial strains.
2. Abstract: it should be one paragraph. The major technical approaches and significant results should be described rather than long background.
3. Introduction: line 51: please don’t present information in the intro as points. The authors should elaborate on the importance of essential oils and their loading on various matrixes such as, biopolymer. The novelty of this study is not clear; please elucidate the novelty in the light of the gap between your study and previous reports.
4. The authors are encouraged to add a schematic illustration, presenting the steps conducted in this study to facilitate the following of the current investigations.
5. Results: Table 1: do you think these biochemical tests are enough? Or you should do other approaches, for instance, VITEK 2. How did you select these isolates among others? Space in table 1 between the genus and species, please correct. The antibacterial results should be presented in details with statistical analysis.
6. Discussion: The authors should discuss the potential antibacterial mechanism of the compounds.
7. Methods: some typos such as line 301 (106). How many replicates did you apply for the antibacterial assays? Did you perform any kind of statistical analysis? Which tests and software did you use?
8. The authors should scrutinize the whole manuscript and write the name of bacterial strains in italic. Moreover, some grammatical mistakes and typos should be corrected.
9. The limitation of this study should be elucidated, highlighting the future perspectives.
Comments on the Quality of English LanguageThe authors should scrutinize the whole manuscript and write the name of bacterial strains in italic. Moreover, some grammatical mistakes and typos should be corrected.
Reviewer 2 Report
Comments and Suggestions for Authors
The introduction provides an update on current reference research on strains of Enterococcus Spp. from broiler chickens.
In Table 2: Why were the five antibiotics chosen?
In figure 2: How can the E. cecorum infection be controlled and when did the chicken get infected?
Line no:250-252: Add a discussion section with more sentence importance on this research.
Please indicate the manuscript with recent citated papers.
Add the missing DOI to the section on references.
Reviewer 3 Report
Comments and Suggestions for Authors
Title: In Vitro Evaluation of Phytobiotics Mixture Antimicrobial Potential Against Enterococcus Spp. Strains Isolated From Broiler Chicken.
The manuscript by Wódz et al. evaluated the antimicrobial potential of phyobiotics mixtures against Enterococcus. Overall, the manuscript is interesting. It requires significant revision before its publication in IJMS as follows:
Comments:
1. Please minimize the typos in the title and main text.
2. Overall, the manuscript lacks novelty or significance. Please justify it adequately.
3. Introduction, too many small paragraphs. Please combine them as final only 2-3.
4. Lines 45-65, please present such information precisely, and the role of these organisms can be well-defined in boiler Chicken, i.e. https://doi.org/10.1016/j.scitotenv.2022.155300.
5. Why not molecular characterization studied to identify strains i.e. 16S RNA? Please provide such information. In addition, please provide brief details on real-time PCR data and analysis.
6. Please illustrate the antimicrobial mechanism and its validation using instrumental analysis i.e. SEM etc.
7. A lot of literature is available on similar studies. Therefore, the discussion requires significant improvement with recent citations to justify the significance of the findings and perspectives.
Comments on the Quality of English LanguageModerate revision requires.
Round 2
Reviewer 1 Report
Comments and Suggestions for Authors
The authors addressed the major comment; however, minor corrections should be considered as follows:
1. Line 47: E.feacalis,E.faecium: please correct.
2. Line 348: please correct.
3. Line 353: E. cecorum, E. faecalis, E. faecium, E. gallinarum, and E. hirae (Please correct.
4. Line 367-368: please correct; for instance, 95 °C for 15 sec.
5. Line 283: number of the reference (Gallucci et al), please add.
5. Line 415: the space, please delete.
6. Figure 6: the number of the figure is missed in the text. I recommend dragging the figure to be after intro if it is possible according to your plan.
7. Line 333: it should be: For this reason, further investigations should be conducted to test the two……….
8. Many spaces should be deleted.
9. The numbers of Table 3, 4 and 5 are missed in the text.
Comments on the Quality of English LanguageMinor editing should be considered; for instance, the name of the organism should be italic and spaces between genus and species.
Reviewer 3 Report
Comments and Suggestions for Authors
Accept.
Comments on the Quality of English Languagenone
Author Response
Best regards, thank you for the review and comments that helped improve the manuscript.